# Topographic enhancement of vertical turbulent mixing in the Southern Ocean

A. Mashayek[1], R. Ferrari[1], S. Merrifield[1], J.R. Ledwell[2], L. St Laurent[2] & A. Naveira Garabato[3]

It is an open question whether turbulent mixing across density surfaces is sufficiently large to play a dominant role in closing the deep branch of the ocean meridional overturning circulation. The diapycnal and isopycnal mixing experiment in the Southern Ocean found the turbulent diffusivity inferred from the vertical spreading of a tracer to be an order of magnitude larger than that inferred from the microstructure profiles at the mean tracer depth of 1,500 m in the Drake Passage. Using a high-resolution ocean model, it is shown that the fast vertical spreading of tracer occurs when it comes in contact with mixing hotspots over rough topography. The sparsity of such hotspots is made up for by enhanced tracer residence time in their vicinity due to diffusion toward weak bottom flows. The increased tracer residence time may explain the large vertical fluxes of heat and salt required to close the abyssal circulation.

[1] Department of Earth, Atmosphere and Planetary Sciences, Massachusetts Institute of Technology, Cambridge, Massachusetts 02139, USA. [2] Woods Hole Oceanographic Institution, Woods Hole, Massachusetts 02543, USA. [3] National Oceanography Centre, University of Southampton, Southampton SO14 3ZH, UK. Correspondence and requests for materials should be addressed to A.M. (email: ali_mash@mit.edu).

Turbulent mixing in the ocean interior plays a leading role in supporting the ocean meridional overturning circulation (MOC) and its associated transports of heat, carbon and biological nutrients. In particular, mixing across density surfaces (also known as diapycnal mixing) is the main process that allows bottom waters to rise across the stable ocean stratification up to a depth of about 2,000 m. Shallower waters are brought to the surface by the westerly winds blowing over the Southern Ocean without any need of diapycnal mixing[1]. While the importance of diapycnal mixing in the deep ocean has been recognized since the seminal work of Munk[2], a quantification of its impact on the MOC and tracer transports remains elusive.

Diapycnal mixing in the deep ocean is primarily the result of breaking internal waves or benthic boundary layer processes occurring at scales from millimetres to tens of metres. The mixing generated by all these turbulent processes is typically quantified in terms of a turbulent diapycnal diffusivity, $\kappa$, which measures the rate at which the turbulence spreads a tracer across density surfaces over time. Munk[2] first estimated that an average turbulent diapycnal diffusivity of $\kappa \sim \mathcal{O}(10^{-4})\,\mathrm{m^2\,s^{-1}}$, a value three orders of magnitudes larger than the molecular diffusivity for heat, is required to allow the observed $\mathcal{O}(10)$ Sv (1 Sv $= 10^6$ $\mathrm{m^3\,s^{-1}}$) flow across the deep ocean density surfaces. Values of $\kappa$ an order of magnitude larger or smaller would imply an MOC much larger or smaller than observed.

A major challenge in directly estimating the average $\kappa$ in the abyssal ocean is the remarkable range of scales involved, from the millimetre scales of mixing to the thousands of kilometres of the MOC. Simultaneous direct measurements over such a large range of scales is beyond present technologies. Oceanographers have therefore resorted to several different approaches over the last five decades to paint a full picture. Direct measurements with turbulent probes deployed along vertical casts, recently reviewed by Waterhouse et al.[3], have found weak turbulent diffusivities of $\mathcal{O}(10^{-5})\,\mathrm{m^2\,s^{-1}}$ in most of the ocean interior, except within a few hundred metres of rough topographic features, where the values are one to two orders of magnitude larger. Tracer release experiments have confirmed that mixing rates are weak in the upper kilometre of the ocean and increase to $\kappa \sim \mathcal{O}(10^{-4})\,\mathrm{m^2\,s^{-1}}$ around rough ocean topography[4–7]. But the tracers appear to experience the large mixing over a much larger area than just within a few hundred metres of the ocean bottom. More complete inverse calculations seem to demand that the large-scale distributions of temperature, salinity, and other tracers experience basin-averaged diapycnal diffusivities of $\mathcal{O}(10^{-4})\,\mathrm{m^2\,s^{-1}}$ below 2,000 m[8–10]. These different lines of evidence are not quite consistent and suggest that we are still lacking a good understanding of how high mixing near rough topographic features affects the MOC and thereby the global distributions of tracers.

The diapycnal and isopycnal mixing experiment in the Southern Ocean (DIMES) was conceived with the explicit goal of investigating the role of topography in setting the distribution of diapycnal and isopycnal mixing in the Antarctic Circumpolar Current (ACC), a key region for the global MOC. The experiment consisted of a release in 2009 of an anthropogenic tracer in the ACC, upstream of Drake Passage, on a neutral density surface at a depth of roughly 1,500 m at the location shown by a star in Fig. 1a. Starting in 2010, the tracer was surveyed at regular intervals over the next several years in the southeastern Pacific and the Scotia Sea, in the region shown in Fig. 1a, and diapycnal diffusivities have been inferred from spreading of the tracer across density surfaces. Diapycnal diffusivities were also inferred from free-fall microstructure profilers that measure the rate of dissipation of turbulent kinetic energy and gave an independent measure of the small-scale turbulence that controls diapycnal mixing.

The microstructure and tracer-based methods both found $\kappa \sim \mathcal{O}(10^{-5})\,\mathrm{m^2\,s^{-1}}$ with small vertical variations upstream of the Drake Passage, where topography is relatively smooth and turbulence is weak[11]. Downstream of the Drake Passage, where major topographic features lead to strong steering of the ACC fronts (Fig. 1a) and more vigorous turbulence and mixing, the two methods have proven more difficult to reconcile. The spreading of tracer across density surfaces implies $\kappa \sim \mathcal{O}(10^{-4})\,\mathrm{m^2\,s^{-1}}$, while the microstructure-based diffusivity is an order of magnitude smaller at the mean depth of the tracer, albeit being up to two orders of magnitudes larger close to topography[12–17].

In this paper, we employ a high-resolution numerical model of the ACC flow in the Drake Passage to reconcile the tracer and microstructure estimates of mixing. More specifically, we inject a numerical tracer in the model to investigate how often it is advected by the mean currents and mesoscale eddies over topographic features where diapycnal mixing is very large. The vertical profile of diffusivity acting on the numerical tracer is prescribed based on DIMES microstructure data. We find that the numerical tracer spreads rapidly across density surfaces, because it is advected over the seamounts and ridges that stick up above the abyssal plains at the tracer depth and spends enough time there to experience an overall $\kappa$ of $\mathcal{O}(10^{-4})\,\mathrm{m^2\,s^{-1}}$.

This study, using a combination of microstructure and tracer observations together with numerical simulations, suggests that in the Drake Passage region, topographically induced diapycnal mixing, together with lateral stirring by mean flows and mesoscale eddies, is sufficiently strong to mix tracers at a rate of $\kappa \sim \mathcal{O}(10^{-4})\,\mathrm{m^2\,s^{-1}}$ across a density surface whose mean depth is close to 1,500 m. In the conclusion, we will argue that a similar picture may apply to the rest of the deep ocean as well, where the long residence time of tracers near mixing hotspots results in large fluxes across density surfaces.

## Results

**Microstructure-based estimates of mixing.** Starting with Osborn[18], oceanographers have inferred mixing rates in the stratified ocean using a relationship between the diapycnal diffusivity, $\kappa$, and the rate of turbulent kinetic energy dissipation, $\epsilon$,

$$\kappa = \Gamma \frac{\epsilon}{N^2}, \qquad (1)$$

where $N^2 = -g\gamma_z/\rho_0$ is the buoyancy frequency, $g$ is the gravitational acceleration, $\rho_0$ is a reference density, and $\gamma_z$ is the vertical gradient of neutral density, that is, the vertical gradient of in-situ density minus the dynamically irrelevant gradient due to compressibility of seawater[19]. The parameter $\Gamma$ is a flux coefficient typically taken to be equal to 0.2. While variations in $\Gamma$ and simplifying assumptions underlying the derivation of (1) add uncertainty to the estimates of $\kappa$ by up to a factor of about three[20,21], this uncertainty is well within the errors of mixing estimates presented below and it is not of leading importance.

As part of the DIMES experiment, a number of vertical profiles of $\epsilon$ were acquired by free-fall microstructure profilers that measured centimeter-scale velocity and temperature fluctuations. We employ 67 microstructure profiles collected during 5 DIMES cruises between 2010 and 2013. The profiles were taken in a sector between the SubAntarctic Front (SAF) and the Polar Front (PF) at locations marked by black stars and circles in Fig. 1b. Measurements along the Phoenix Ridge and the Shackleton Fracture Zone were collected during the US2 and US5 cruises[16,22]. Measurements in the eastern part of the sector shown in Fig. 1b were collected during cruises UK2.5, UK3, and UK4, along the same transect as the approximately meridional line near 57W, known as the World Ocean Circulation

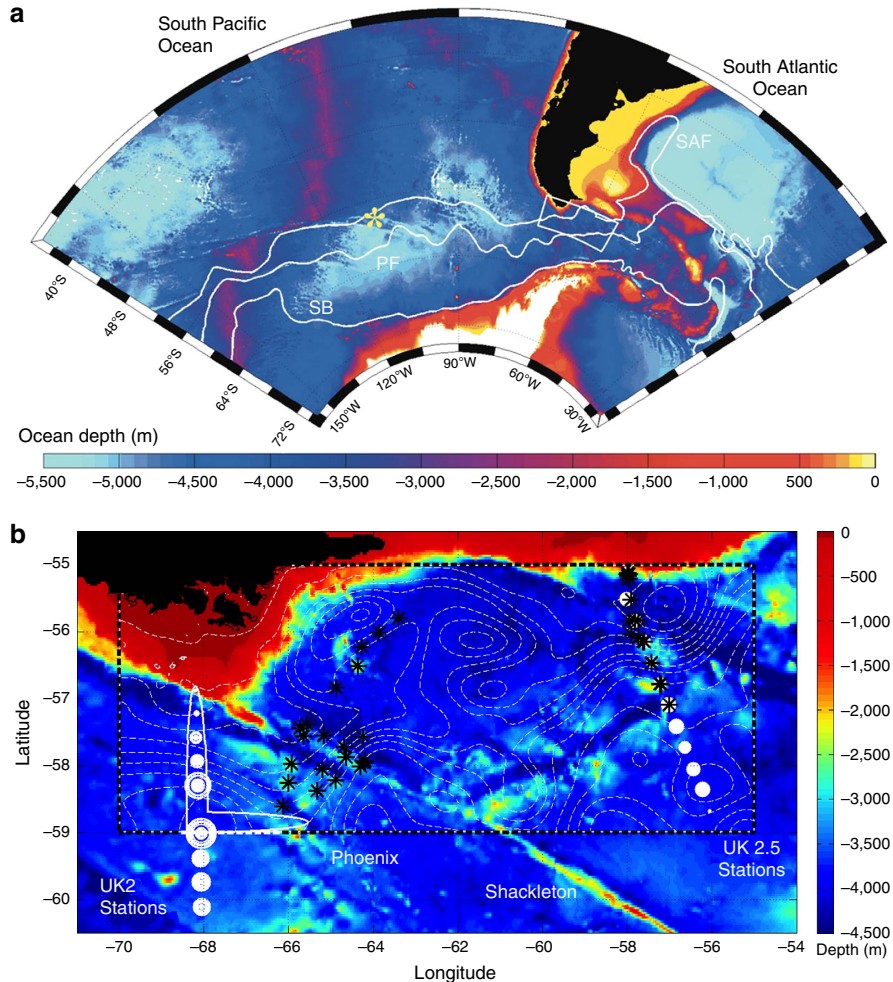

**Figure 1 | Locations where vertical profiles of microstructure and tracer were collected overlain on the topography of the Drake Passage region in the Southern Ocean.** (**a**) Bathymetric map showing the Drake Passage region in the Southern Ocean[27]. The area covers the domain of analysis of DIMES. The white rectangle encloses the domain used for the high-resolution numerical simulation. The yellow star represents the location of release of an anthropogenic tracer in DIMES. The white lines represent three of the major Antarctic Circumpolar Current system's fronts, namely the SubAntarctic Front (SAF), the Polar Front (PF) and the Southern Boundary front (SB). (**b**) An enlarged view of the white box in **a**. White circles represent sampling stations of DIMES tracer along UK2-68W and UK2.5-SR1 cruise tracks, with the circle radii proportional to the vertically integrated tracer concentrations. Black stars represent DIMES microstructure measurements during US5 and UK2.5 cruises. The half ellipses represent the location where the numerical tracer was injected in the high-resolution simulation. White dashed contour lines represent sea surface height.

Experiment (WOCE) Scotia Ridge SR1b transect[15,23] (see DIMES homepage at http://dimes.ucsd.edu/en/ for details of cruises).

Mean vertical profiles of buoyancy frequency $N^2$ (Fig. 2a), $\epsilon$ (Fig. 2b) and $\kappa$ (Fig. 2c) were constructed from the 67 microstructure profiles by averaging the measurements over 100 m bins using a height above bottom ($h_{ab}$) coordinate. The $h_{ab}$ coordinate is used to capture the bottom enhancement of $\epsilon$ and $\kappa$ due to turbulence induced by interactions of bottom flows with topography. The majority of profiles ended within $50 \pm 25$ m of the bottom. To isolate the surface mixed layer and region of strong T/S interleaving, data shallower than 1,000 m were not included in this analysis. A 500 m running mean was applied to the resulting average profiles to reduce high frequency fluctuations. The standard error was computed using a bootstrap method treating each profile as an independent sample. Detailed cruise information as well as a more detailed description of the methodology are provided in St. Laurent *et al.*[16] and Merrifield *et al.*[22].

Our goal is to test whether the mixing profiles measured with the microstructure profilers are consistent with the overall mixing sampled by a tracer released in the same region. To this end, we constructed a three-dimensional (3D) $\kappa$ map (to be used in a numerical model) by imposing the mean diffusivity profile in Fig. 2a over the entire numerical domain as a function of $h_{ab}$. Thus, seamounts and ridges will result in large $\kappa$ values further up in the water column than in deep valleys and trenches. Assuming that the profile of $\kappa$ is only a function of $h_{ab}$ may seem naive, but it is supported by the measurements. Panel (b) in Fig. 2 shows profiles of $\epsilon$, $N^2$ and $\kappa$ as a function of $h_{ab}$ averaged on casts over topography deeper and shallower than 2,500 m, respectively. Both $\epsilon$ and $N^2$ are larger over shallower topography, but their ratio $\kappa$ is very similar, if plotted as a function of $h_{ab}$. Thus the mean of all profiles seems to be representative of the entire domain of our focus.

By imposing a 3D $\kappa$ map that is only a function of $h_{ab}$, we ignored regional variations in the profile of $\kappa$. For example, it appears that $\kappa$ is larger where bottom velocities are larger[16], but sampling during the DIMES cruises is too coarse to quantify these variations. The model success in reproducing the evolution of the tracer with a $\kappa$ profile only function of $h_{ab}$, as shown next,

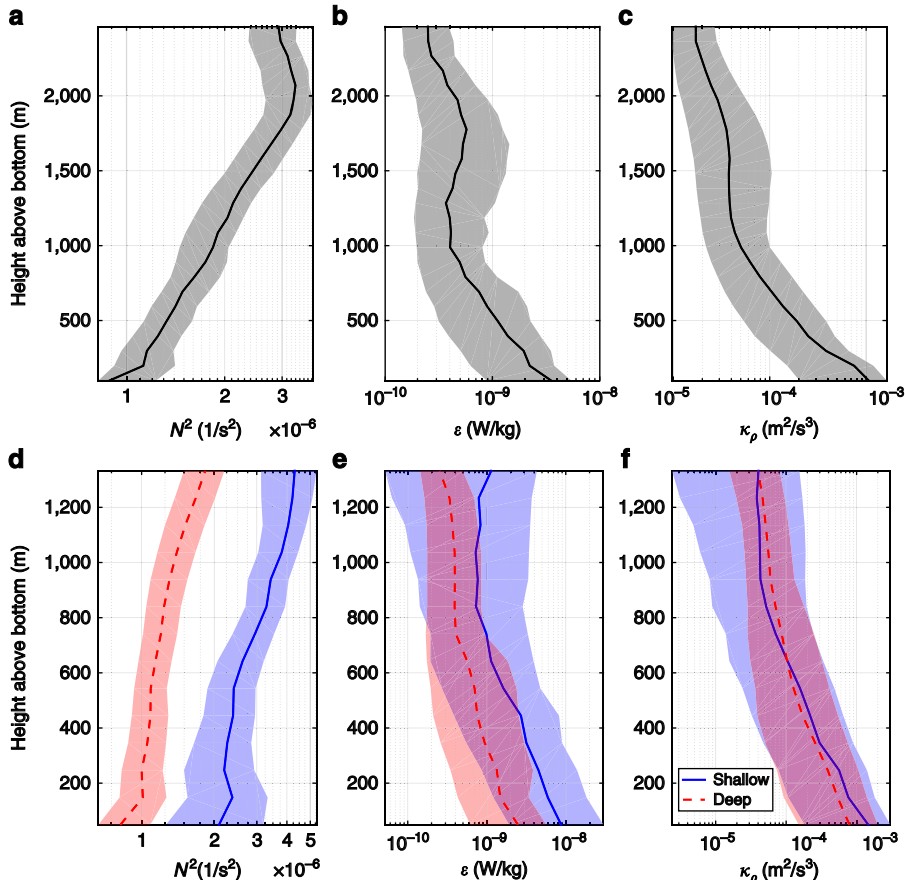

**Figure 2 | Profiles of stratification and dissipation of turbulent kinetic energy and diffusivity.** (**a–c**) Mean profiles of buoyancy frequency, $N^2$, rate of dissipation of kinetic energy, $\epsilon$, and effective diapycnal diffusivity, $\kappa$, plotted as a function of height above bottom, $h_{ab}$. The profiles are constructed from all the 67 microstructure profiles shown by stars in Fig. 1b. The shading represents standard error. (**d,e**) Same as top row, but with profiles divided into a group of 12 profiles shallower than 2,500 m (blue shading and continuous lines) and a group of 45 profiles extending deeper (red shading and dashed lines).

suggests that regional variations are not of leading order importance for our study.

**Tracer-based estimates of mixing.** The microstructure profiles present a largely under-sampled view of a highly heterogeneous mixing field in the Drake Passage region. To obtain a measure of the spatiotemporal averaged mixing in the same region, an anthropogenic tracer was released in the South Pacific Ocean as a part of DIMES. The passive chemical trifluoromethyl sulphur pentafluoride ($CF_3SF_5$) was released in February of 2009 along the neutral density surface of 27.9 kg m$^{-3}$ at the location marked as a yellow star in Fig. 1a, 2,000 km west of the Drake Passage, between the Polar Front and the SubAntarctic Front. In the subsequent two years, the tracer was sampled at various stations downstream of the release point. As discussed earlier, our focus is on the region between the Drake Passage and the Western Scotia Sea. More specifically, we focus on the area between the 68 W transect of the UK2 cruise and the 57 W transect (a.k.a SR1) of the UK2.5 cruise. The UK2 cruise also included transects at 79 W and 57 W, while the UK2.5 cruise also included a transect at 78 W. We will not be concerned with these additional transects, two of which lie outside our computational domain. We will only consider UK2-68W and UK2.5-57W transects and in short will only refer to them as UK2 and UK2.5 stations.

Figure 3a shows measured tracer profiles as a function of density from the UK2 stations at the western end of the sector shown in Fig. 1b. The tracer concentrations peak at the original target release density. The thick red line represents a mean over all profiles at UK2 stations. In Fig. 3d, we show the same profiles using depth as the vertical coordinate. The conversion from density to depth coordinates is based on the mean depth of each density surface averaging over all profiles and subtracting the mean depth of the target density. Fig. 3b,e are similar to Fig. 3a,d but for the UK2.5 stations in the East Scotia Sea.

The mean profiles in Fig. 3d,e are equivalent to Fig. 2b,f in Watson *et al.*[17], who used them to estimate the diapycnal diffusivity experienced by the tracer between the UK2 and UK2.5 sections. To do so, they solved the advection–diffusion equation for the tracer on a longitude-depth 2D domain, adjusting the vertical and horizontal diffusion and the lateral advective velocities until they found the best fit to the mean profiles in Fig. 3d,e. This approach returned a best estimate of the vertical diffusivity of $3.6 \pm 0.6 \times 10^{-4}$ m$^2$ s$^{-1}$. The centre of mass of the tracer between UK2 and UK2.5 stations was at a depth of $\sim 1,400$ m, equivalent to a height above bottom $h_{ab} \sim 2,200$ m. The mean diapycnal diffusivity at $h_{ab} \sim 2,200$ m based on the microstructure profiles presented in Fig. 2c is $\kappa \mid_{h_{ab}=2,000m} \sim 3 \times 10^{-5}$ m$^2$ s$^{-1}$, a full order of magnitude smaller than the bulk value inferred by Watson *et al.*[17] from vertical dispersion of the tracer. There appears to be an order of magnitude discrepancy between the mixing actually measured by the microstructure profilers and the mixing experienced by the tracer.

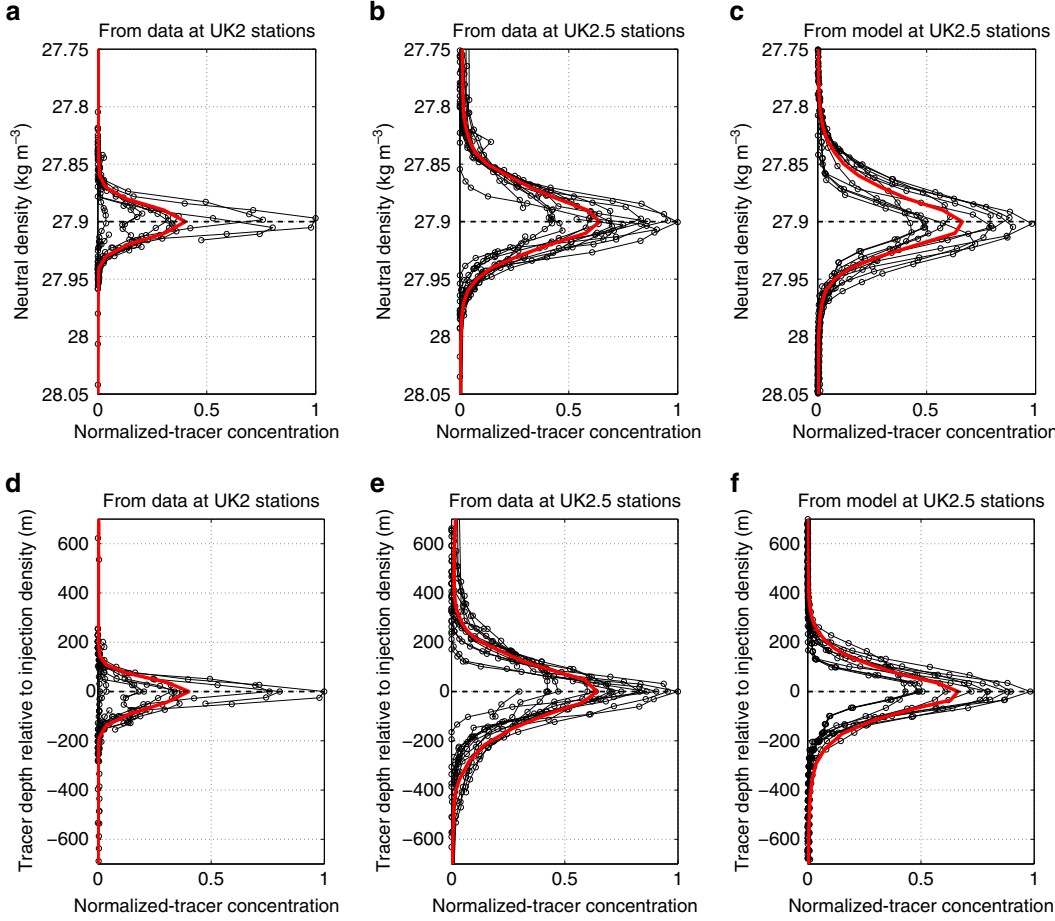

**Figure 3 | Vertical profiles of tracer concentration upstream and downstream of the Drake Passage.** Measured tracer concentrations from observations at the UK2-68W stations (**a,d**) and UK2.5-SR1 stations (**b,e**) as well as from the model simulation results sampled at the locations of the UK2.5-SR1 stations (**c,f**). Concentrations in each panel are normalized by maximum concentration over their corresponding set of stations. The black lines are individual profiles at each station and the red lines are the averages over all individual profiles. (**a–c**) The profiles as a function of neutral density, while panels (**d–f**) show the same profiles as a function of depth. The mean profiles are mapped from density onto depth using the mean depth of isopycnals at each set of stations. The model tracer is initialized in density space with the mean observed profile at UK2 stations (red line in left-top panel). Time difference between UK2-68W and UK2.5-SR1 measurements was 120 days. For comparison, the model profiles are plotted in **c,f** 120 days after release along UK2-68W.

The comparison between the two estimates, however, is not straightforward: while the centre of mass of the tracer travels substantially above any topography, the density surfaces occupied by the tracer episodically come close to the seafloor and experience enhanced mixing. The bottom values of diffusivity are up to two orders of magnitude larger than the mid-depth values and can substantially increase the mean mixing experienced by the tracer. To quantify the increase in mean diffusivity resulting from the intermittent advection of tracer toward boundary-enhanced mixing regions, we turn to a high-resolution simulation of advection and diffusion of the tracer over the domain shown in Fig. 1b.

**Numerical model**. We employ a numerical model to investigate whether the lateral transport by the ocean velocity field brings the tracer in sufficient contact with high mixing rough topography to explain why the average diapycnal diffusivity experienced by the tracer is $\sim \mathcal{O}(10^{-4})\,\mathrm{m^2\,s^{-1}}$. To answer this question, the model must accurately reproduce the ocean velocity field in the Drake Passage region and resolve the topographic features that influence mid-depth mixing.

Tulloch et al.[24] ran a numerical simulation of a Southern Ocean sector spanning 140 longitude degrees and 40 latitude degrees, centred on the Drake Passage region—the whole region shown in Fig. 1a—with a horizontal resolution of 1/20th degree ($\sim 3\,\mathrm{km} \times 6\,\mathrm{km}$) and 100 vertical levels of unequal thickness such that the top 70 layers, which span the top 1,900 m, are all $< 35\,\mathrm{m}$ thick. The model was forced at the open boundaries by restoring velocity, temperature and salinity to the Ocean Comprehensive Atlas (OCCA), which is an 18-month-long ocean state estimate[25]. They showed that the model reproduced mean velocity and stratification in close agreement with WOCE climatology[26] and mesoscale eddy variability in agreement with satellite altimetry and mooring observations. Here, we nest a smaller patch centred over the Drake Passage in the Tulloch et al.[24] domain with a higher horizontal resolution of 1/100th of degree ($\sim 600\,\mathrm{m} \times 1\,\mathrm{km}$), but the same vertical resolution. This nested domain is shown as a small white rectangle in Fig. 1a and as the whole region in Fig. 1b. The higher horizontal resolution is necessary to fully resolve the bathymetric features in the Smith and Sandwell one minute (1/60th of degree) product[27]. The domain spans the longitudinal band between UK2 and UK2.5 cruises, because strong diapycnal spreading of tracer has been

diagnosed in this sector[17]. The nested patch is restored toward the Tulloch *et al.* simulation within a strip 1/10th of degree wide along the open boundaries on a timescale of 4 days. We verified that the model overestimates the eddy kinetic energy levels measured by AVISO by 50% and reproduces the ~1,500 m vertical decay scale of the kinetic energy as compared with the Drake Passage mooring data presented in Tulloch *et al.*[24] where more details on the numerical model are also presented.

Our goal is to study how a tracer stirred along density surfaces by the mean currents and the mesoscale eddy field in the Drake Passage is mixed across density surfaces by turbulent mixing. This is achieved by releasing a tracer in the numerical model along the 27.9 kg m$^{-3}$ density surface. The tracer is initialized with tracer concentration values measured at the UK2 stations. The vertically integrated tracer concentrations, given by the size of the white dots in Fig. 1b, approximately follow a Gaussian distribution in latitude. We thus fit a Gaussian to those profiles and use it to initialize the numerical tracer. For the few UK2 stations that were taken south of the numerical domain, we follow the mean ACC streamlines from the stations to the southern edge of the domain and apply the tracer concentrations there. The vertical distribution of the numerical tracer is also Gaussian about the target density of 27.9 kg m$^{-3}$ and is prescribed in density space to match the UK2 mean vertical tracer profile shown by the red line in Fig. 3a.

In addition to being advected along density surfaces by the velocity field generated by the model, the tracer is also mixed in the vertical with the 3D map of $\kappa$ generated from the microstructure profiles. Figure 4a shows the high-resolution bathymetry in our model along with the 3D $\kappa$ map. We apply the same vertical profile of $\kappa$ as a function of $h_{ab}$ at each location. By design, major topographic features will be hotspots of mixing at the depth of the tracer release experiment. We verified that the spurious numerical diffusion in the model is below $10^{-5}$ m$^2$ s$^{-1}$, which is safely smaller than the range of diffusivity we are concerned with in this work. Thus, we can employ the model to study spreading of the tracer subjected to the spatially variable mixing map in Fig. 4a.

Figure 4b shows a snapshot of the numerical tracer patch 150 days after release. The strong eddy field that develops in the model as a result of baroclinic instabilities rapidly stirs the tracer over the whole domain, thereby getting some fraction of the tracer to come in contact with the topographic features that reach

the tracer depth. This fraction experiences larger mixing rates as illustrated by colour coding of the numerical tracer.

To illustrate the skill of the model in reproducing the measured spreading of tracer across density surfaces, we compare the vertical tracer profiles measured at the UK2.5 stations and shown in Fig. 3b, with the numerical tracer concentrations simulated by the model at the same locations, 120 days after the tracer was released in Fig. 3c—120 days is approximately the mean time it takes the model tracer to cross the distance between the UK2 and UK2.5 stations and also the approximate time lapsed between the actual UK2 and UK2.5 cruises. The agreement between the numerical and the observed profiles attest to the skill of the model in integrating discrete observations into a continuous and dynamically consistent framework, which we will next use to quantify mixing in the Drake Passage. Given that Watson *et al.*[17] inferred their diffusivity from the mean vertical tracer profiles at the UK2 and UK2.5 stations, the close comparison between the observed and simulated tracer profiles also implies that the model reproduced the observational result that the tracer experienced a $\kappa \simeq 3.6 \times 10^{-4}$ m$^2$ s$^{-1}$. And this value of $\kappa$ was generated by mixing the tracer with a profile of diapycnal diffusivity based on microstructure data.

**Analysis of the model output**. We define a tracer-weighted diffusivity for the numerical tracer at a particular instant in time as the average of the diapycnal diffusivity weighted by the tracer concentration:

$$\bar{\kappa} = \frac{\iiint \kappa \, c \, \mathrm{d}V}{\iiint c \, \mathrm{d}V}, \qquad (2)$$

where the volume integral is taken over the whole domain. The time evolution of $\bar{\kappa}$ is shown in Fig. 5 as a solid blue line. $\bar{\kappa}$ increases rapidly over the first tens of days as the tracer is spread laterally by the geostrophic eddy field and comes into contact with shallow rough topographic features where the diapycnal diffusivity is very large. The initial transient is marked with the leftmost grey shading zone in the figure. After this transient the average diffusivity settles to approximately $3 \times 10^{-4}$ m$^2$ s$^{-1}$, a value consistent, within uncertainty, with that inferred in the DIMES experiment from the tracer profiles at the UK2 and UK2.5 stations (dashed-dotted horizontal line in Fig. 5) and discussed in relation to Fig. 3d[17]. This suggests that there must be

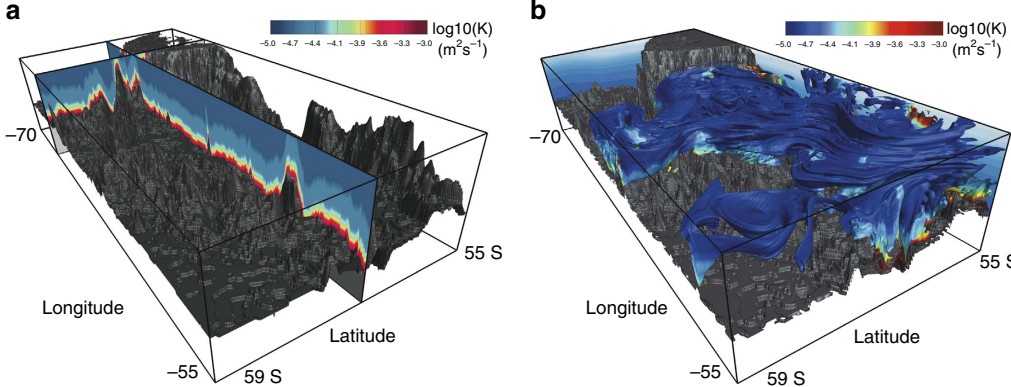

**Figure 4 | Sections of diapycnal diffusivity map used in the numerical model and snapshot of the numerical tracer.** (**a**) The same mean diffusivity profile in Fig. 2f is imposed everywhere in the domain as a function of height above the bottom. Two orthogonal sections through the resulting 3D map of diffusivity are shown in colour to illustrate the horizontal and vertical variations in diffusivity which arises due to changes in bathymetry. (**b**) A snapshot of tracer distribution at 150 days into the simulation. The colour represents the strength of the diffusivity the tracer experiences, with red highlighting the large diffusivities close to the seafloor (see **a** and Fig. 2). While the red regions are very rare, they dominate the net mixing experienced by the tracer. The eastward ACC flow advects the tracer from the back to the front of the figure, while it is also stirred by eddies along the way. The contours on the western and northern faces represent density surfaces which shoal towards Antarctica, that is, towards the southern (left) boundary of the domain.

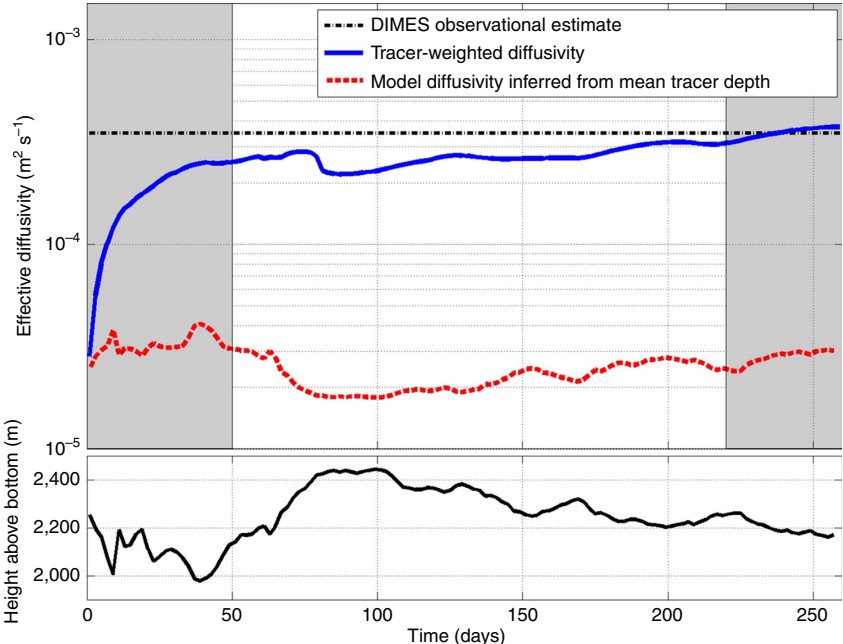

**Figure 5 | Different diagnostics of diffusivity from the numerical tracer.** The top panel shows two different estimates of diapycnal diffusivity diagnosed from the model tracer release experiment. The dashed-dotted horizontal line represents the diffusivity inferred from DIMES observations of vertical dispersion of the tracer in the Drake Passage between UK2-68W and UK2.5-SR1 stations[17]. The dashed red line is the value of the diffusivity profile shown in Fig. 2f at the mean height above the bottom of the tracer as a function of time. The solid blue line represents the tracer-weighted diffusivity computed with equation (2) as a function of time. The left grey shading highlights the spinup time during which eddies stir the tracer and bring it in contact with bottom roughness as it enters Drake Passage. The right grey shading highlights marks the time in which more than 25% of tracer has left the computational domain. The bottom panel shows the temporal evolution of the mean height above bottom ($h_{ab}$) of the model tracer.

enough tracer close to tall topographic ridges and seamounts, where mixing is very strong, that it drives the average diapycnal diffusivity to $\mathcal{O}(10^{-4})\,\mathrm{m^2\,s^{-1}}$. The small positive trend in $\bar{\kappa}$ after day 50 is due to advection of tracer out of the downstream end of the domain. Tracer far from any topographic feature is advected faster out of the domain than tracer closer to topographic seamounts and ridges, as we explain below. Thus the fraction of tracer experiencing strong mixing close to topographic features increases over time. We restrict our analysis to the time period between day 50 and 220, when the increase in $\bar{\kappa}$ is small. After day 220, indicated as the rightmost grey shading region in Fig. 5, more than 25% of the tracer has left the domain and the increase in $\bar{\kappa}$ becomes significant.

The choice of weighting the diapycnal diffusivity by the tracer concentration $c$ in equation (2) is somewhat arbitrary. One could argue that weighting by the vertical gradient squared, $c_z^2$, is more appropriate, because mixing matters only where there are vertical gradients to act on. Weighing by different powers of the tracer concentration, like $c^2$ would also be reasonable. Here, we stick with what is arguably the simplest choice. In the 'Methods' section (Fig. 9), we show that one gets essentially the same evolution of $\bar{\kappa}$ (within a factor of two) with any of these alternative weighting choices.

It is somewhat a surprise that the numerical tracer-based estimate $\bar{\kappa}$ and the real tracer-based estimate $\kappa$ are so close, because they are based on different metrics. $\bar{\kappa}$ represents the diapycnal diffusivity experienced by the overall numerical tracer at an instant in time. The observational estimate, instead, is the result of mixing experienced by the tracer as it was advected and dispersed by eddy stirring from the UK2 to the UK2.5 transects and thus includes both time and spatial averaging. It is indeed possible that the agreement is somewhat fortuitous. But what matters here is that both the numerical and the real tracer-based

estimates of the diapycnal diffusivity are of $\mathcal{O}(10^{-4})\,\mathrm{m^2\,s^{-1}}$, an order of magnitude larger than the average diapycnal diffusivity estimated from microstructure profiles at the depth where the tracer was injected. Thus our approach of prescribing the $\kappa$-profile based on microstructure data and the naive estimate $\bar{\kappa}$ is sufficient to capture the strong mixing experienced by the tracer.

Encouraged by this comparison, we now use the model output to understand the seeming discrepancy between the microstructure-based $\kappa$ at mean tracer depth and the $\kappa$ sampled by the tracer. To this end, we first calculate the height above the bottom of the centre of mass of the numerical tracer as a function of time. As shown in the bottom panel of Fig. 5, the centre of mass sits between 2,000 m and 2,400 m above the bottom. The imposed microstructure-based diffusivity at this depth above the bottom, $\kappa(h_{ab})$, is shown in Fig. 6b and is close to $2 \times 10^{-5}\,\mathrm{m^2\,s^{-1}}$. The value of $\kappa(h_{ab})$ at the depth of the centre of mass of the tracer as a function of time is shown as a red-dashed line in the top panel of Fig. 5. This value is an order of magnitude smaller than the diapycnal diffusivity $\bar{\kappa}$ experienced by the total numerical tracer—the solid blue line computed using equation (2). The source of discrepancy can be identified by comparing the diapycnal diffusivity experienced by the portion of the tracer found within 1 km of the ocean bottom with the portion found more than 1 km above topography. These are computed using equation (2), but restricting the integrals in the numerator and denominator to the volume where the tracer is within and beyond 1 km above the ocean bottom. We found that 150 days into the simulation (the results change little for other times between 50 and 220 days), the portion of the tracer within a kilometre of the ocean bottom experienced an averaged diapycnal diffusivity of $\sim 1.6 \times 10^{-3}\,\mathrm{m^2\,s^{-1}}$, while the tracer above $h_{ab} = 1,000\,\mathrm{m}$ experienced an averaged diapycnal diffusivity more than an order of magnitude smaller $\sim 4 \times 10^{-5}\,\mathrm{m^2\,s^{-1}}$. The overall

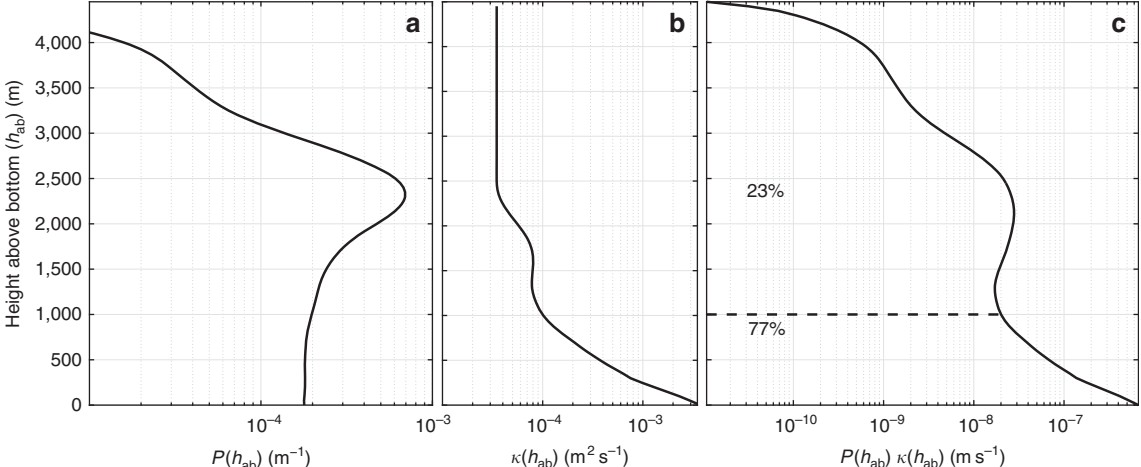

**Figure 6 | Quantification of contribution of bottom enhanced mixing to net diffusivity experienced by the tracer.** (**a**) Probability density function of the tracer, or equivalently concentration of the tracer as a function of height above bottom ($h_{ab}$) on simulation day 150 (same used for Fig. 5) when most of the tracer was still within the model domain. (**b**) The prescribed microstructure-based diffusivity as function of $h_{ab}$. (**c**) Product of the previous two curves. The area between the curve and the vertical axis represents the net diffusivity experienced by the tracer. The little amount of tracer within 1,000 m of the seafloor contributes 77% to the net diffusivity, while the larger amount of tracer further up in the water column contributes only 23%.

diapycnal diffusivity $\bar{\kappa}$ is dominated by the strong mixing acting on the tracer whenever it encounters shallow topographic features.

The importance of strong mixing close to the topography is better quantified by studying the distribution of tracer as a function of depth above the bottom $h_{ab}$. The probability that tracer is found at a certain $h_{ab}$ is given by the function,

$$P(h_{ab}) \equiv \frac{d}{dh_{ab}} \left( \frac{\iiint_{z+H<h_{ab}} c \, dV}{\iiint c \, dV} \right) \qquad (3)$$

where $H$ is the ocean depth, the integral in the numerator is restricted to tracer deeper than $h_{ab}$, while the integral in the denominator is taken over the total volume of the simulation and it remains constant until the tracer leaves the integration domain. The probability function $P(h_{ab})$, plotted in Fig. 6a, peaks at $h_{ab} \simeq 2,300$ m, confirming that most of the tracer is quite far from the seafloor. But there is a non-negligible fraction of tracer within 1 km of the ocean bottom. Figure 6b shows the prescribed $\kappa$ as a function of $h_{ab}$. While most of the tracer sits far above the bottom, where $\kappa(h_{ab})$ values are $O(10^{-5}) \, m^2 \, s^{-1}$, the small fraction of tracer within a kilometre of the ocean bottom experiences $\kappa(h_{ab})$ values one to two orders of magnitude larger. The overall $\bar{\kappa}$ is given by the product of $P(h_{ab}) \times \kappa(h_{ab})$ integrated over all $h_{ab}$. Determining whether the integral of this product is dominated by contributions within the bottom kilometre or above is tantamount to asking whether the tracer comes in sufficient contact with pronounced rough topography to experience a net large average diapycnal diffusivity or not. This is the key question posed in this work.

The product $\kappa(h_{ab}) \times P(h_{ab})$ is shown as a black line in Fig. 6c. Its integral from bottom up is dominated by values below 1 km where the product grows to be two orders of magnitude larger than above. Even though only 7% of the tracer is found within 1 km of the ocean bottom, this portion of the tracer contributes 77% of the integral of $P(h_{ab}) \times \kappa(h_{ab})$. This confirms that even though the amount of tracer decays strongly towards the ocean floor (Fig. 6a), the exponential increase in $\kappa$ towards the bottom (Fig. 6b) makes up for this decay and results in a large overall $\bar{\kappa}$ (Fig. 6c). The relatively high values of $P(h_{ab})$ near the bottom compared with those an equal distance above the peak are largely

due to trapping of tracer near the bottom, as will be discussed below

A possible interpretation of our result is that the tracer experiences enhanced mixing, because there are enough topographic features that extend all the way up to the mid-depth at which the tracer was released. This would be consistent with the traditional explanation that the area-averaged diapycnal diffusivity in the ocean is dominated by mixing hotspots close to topography. But we can show that this is not the case. The dashed red line in Fig. 7a shows that the average diffusivity at the mean tracer depth of 1,500 m is close to $2 \times 10^{-5} \, m^2 \, s^{-1}$. Apparently there are not enough topographic features at mid-depths to lead to a substantial increase in the average diffusivity at a fixed depth. The result does not change much if the average of $\kappa$ is taken along density surfaces—shown as a black line in Fig. 7.

The order of magnitude discrepancy between the area-averaged diffusivity at fixed depth/density and the tracer-weighted average $\bar{\kappa}$ suggests that there must be a tendency for the tracer to accumulate around topographic features and experience large mixing there. This is confirmed by Fig. 8a where we plot the vertically integrated tracer concentration at 150 days into the simulation (same day as for Figs 5 and 7) mapped onto bottom topography. The tracer concentration is larger over topographic features that stick up to mid-depths and come in contact with the tracer. This same effect is evidenced by the tail of higher tracer concentrations close to topography than away from it in Fig. 6a. The accumulation of tracer around tall topographic features appears to result from two effects which together act to increase the residence time of tracer there. First, the increase in $\kappa$ toward the seafloor results in large fluxes of tracer toward tall topographic features. Second, the flow speed decreases close to the seafloor as shown in Fig. 8b. Hence, tracer is efficiently diffused over topographic seamounts and ridges and then remains trapped there for a large time.

To further quantify the impact of higher tracer concentrations close to seamounts and ridges, in Fig. 9 we compare the fraction of tracer within 1,000 m of the seafloor (referred to as the SML, the stratified mixing layer where turbulence is strong but not so strong as to erase stratification) with the fraction of volume occupied by tracer in the SML. Comparison of the two curves in the plot (focusing on the left axis) shows that there is an order of

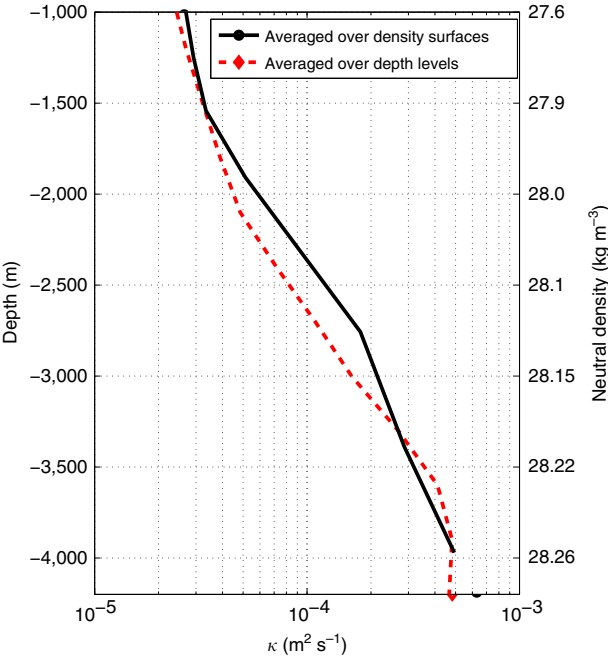

**Figure 7 | Vertical profiles of diffusivity averaged along surfaces of constant depth and density.** Domain-averaged diapycnal diffusivity as a function of depth (dashed red line) and density (solid black line) at the simulation time 150 days (same day as that in Figs 5 and 7). The transformation from depth (left axis) to density (right axis) is done by calculating the mean depth of isopycnals.

magnitude more tracer concentration within 1,000 m of topography than would be the case if the tracer were uniformly distributed over the whole volume it occupies. On the right axis we show the value of the tracer and volume fractions, but multiplied by the average diffusivity in the SML of $4.1 \times 10^{-4}$ $m^2 s^{-1}$. Consistent with what illustrated in Fig. 5, we find that it is the large fraction of tracer within 1,000 m of the seafloor that gives a tracer-averaged diffusivity to $\mathcal{O}(10^{-4})\, m^2 s^{-1}$. If the tracer were uniformly distributed in space, then the volume fraction suggests that the averaged diffusivity would be $\mathcal{O}(10^{-5})\, m^2 s^{-1}$, the same magnitude obtained by averaging $\kappa$ along a density surface in Fig. 7.

In summary, the large diffusivity experienced by the tracer is due to a combination of (i) efficient stirring of the tracer over the whole domain by mesoscale eddies bringing the tracer in contact with rough topographic features, and (ii) long residence time of the tracer around these features, which leads to high tracer concentrations in regions of strong mixing. While the first point has been made previously in the literature, the second point does not seem to have been fully appreciated.

Our conclusion that the enhancement of mixing close to the ocean bottom dominates the spreading of the tracer resonates with an argument first proposed by Armi[28] and subsequently further explored by others[29–31], except for a crucial difference. Armi argued that high mixing in the ocean bottom boundary layers explains the $\mathcal{O}(10^{-4})\, m^2 s^{-1}$ basin-averaged mixing rates in the abyssal ocean. Garrett[32] rebutted that the bottom boundary layers are weakly stratified and thus vigorous overturning of the unstratified fluid in the boundary layer would not lead to enhancement of mixing. The crucial difference in our argument is that (1) the high mixing measured by the microstructure probes occurs in the stratified ocean interior, above the weakly stratified bottom boundary layer but within a kilometre of the ocean

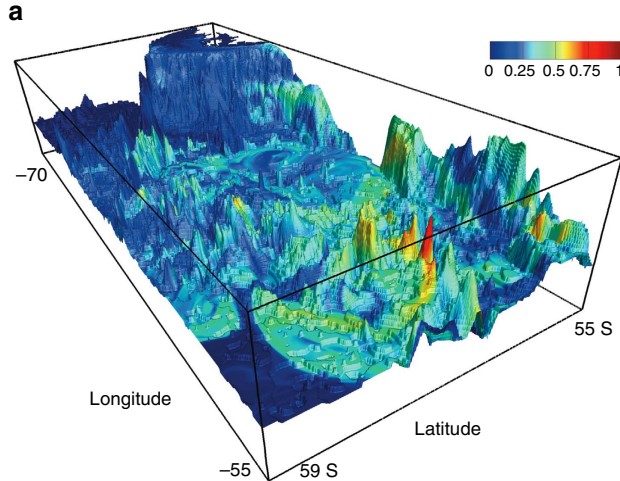

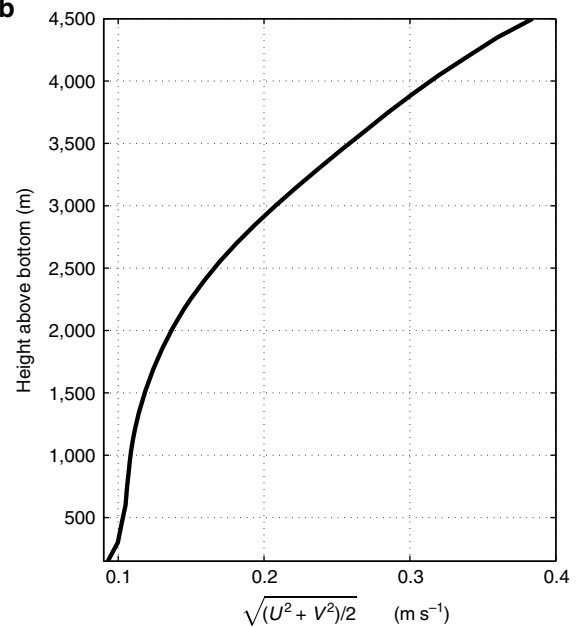

**Figure 8 | Vertically integrated concentration of numerical tracer overlain over model topography and vertical profile of the averaged horizontal velocity.** (**a**) Vertically integrated tracer mapped onto bottom topography. The map is normalized by the maximum vertically integrated concentration in the domain. The plot is made for day 150 day into the simulation, same as Figs 6 and 7. (**b**) Domain-averaged velocity as a function of height above bottom. It was verified that the average does not change significantly when restricted to the volume occupied by the tracer.

bottom and (2) the large residence time of tracers around topography is key.

## Discussion

We used microstructure profiles collected as part of the DIMES experiment to illustrate that diapycnal mixing in the Drake Passage is very heterogeneous, being one to two orders of magnitude larger within a kilometre of topographic features than over deep bathymetry. This heterogeneity was used to explain why measurements of diapycnal diffusivity inferred from the vertical dispersion of a passive tracer at $\sim 1,400$ m was of $\mathcal{O}(10^{-4})\, m^2 s^{-1}$, while concurrent measurements based on

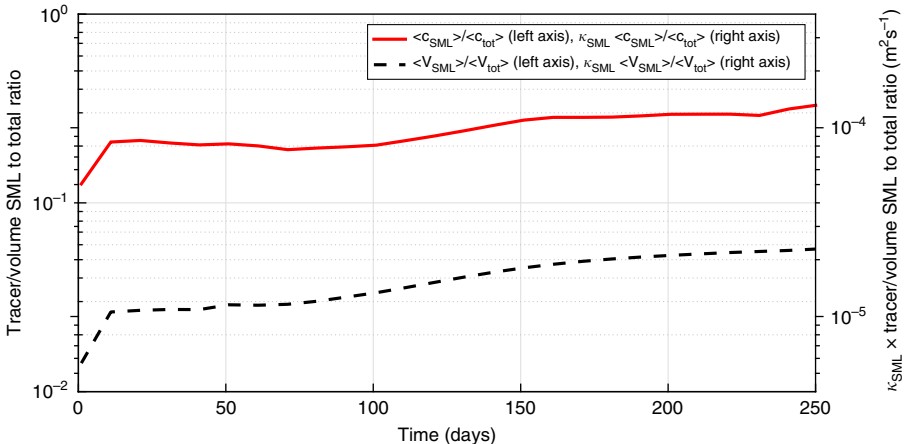

**Figure 9 | Contribution of the numerical tracer close to topography to the net diffusivity experienced by the tracer.** The solid red line represents the fraction of tracer within 1,000 m of the seafloor (referred to as SML) and the dashed line represents the fraction of volume occupied by tracer that is in the SML, that is, the ratio of the volume occupied by tracer within the SML to the total volume occupied by the tracer. The right axis shows the two ratios multiplied by the mean diffusivity within the SML ($\kappa_{SML} \simeq 4.1 \times 10^{-4} \text{ m}^2 \text{ s}^{-1}$). The plot is made for day 150 into the simulation, same day used for Figs 6,7 and 8.

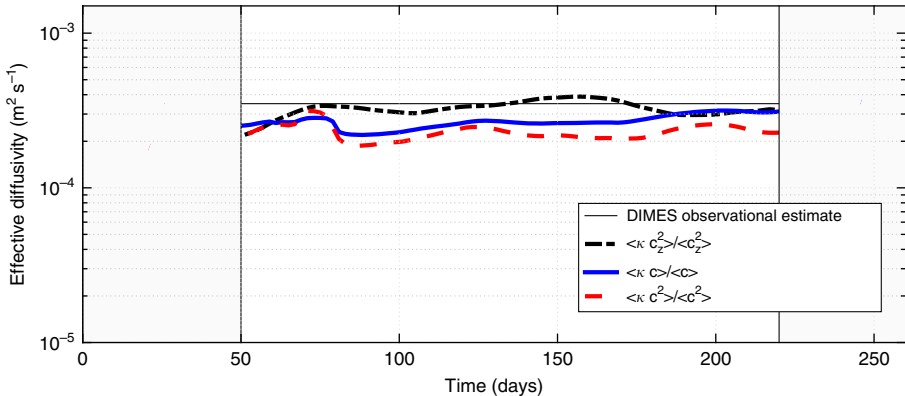

**Figure 10 | Different estimates of the net diffusivity experienced by the numerical tracer.** Comparison of different tracer-weighted mean diffusivities, using $c$, $c^2$ and $c_z^2$ as weighting factors, that is, replacing $c$ with the different weighting factors in equation (2). The thick solid blue line is the same as that shown in Fig. 5.

microstructure profiles of dissipation of turbulent kinetic energy found much smaller values of $\mathcal{O}(10^{-5}) \text{ m}^2 \text{ s}^{-1}$ at the same depth. A numerical model was used to follow the evolution of the tracer as it was advected by the strong jets and geostrophic eddies that characterize the horizontal circulation in the Southern Ocean. The tracer experienced both the very high mixing above shallow topographic features and the much weaker mixing over deeper bathymetry. Thus the diapycnal diffusivity experienced by the tracer was the average of strong mixing over shallow hotspots and weak mixing elsewhere.

Our work offers strong evidence that diapycnal mixing of tracers at mid-depths in the Drake Passage is enhanced because stirring by geostrophic eddies and mean flows brings tracers in contact with shallow seamounts and ridges, where diapycnal diffusivities are one to two orders of magnitude larger than background values. The model further suggests that the large residence time of tracers around topographic features, because of slow mean flows and topographically locked recirculating eddies, is crucial in explaining the large diapycnal diffusivity experienced by the tracer. There is an extensive literature suggesting that the large diapycnal diffusivities inferred from tracer distributions in the deep ocean must be the result of strong mixing at localized hotspots, like rough topography[33]. However, Fig. 7 shows that in

the Drake Passage the mixing hotspots are too rare to significantly affect the area-averaged diffusivities over the whole passage. In the model it is the disproportionately long time spent by the tracer close to rough topographic features, where mixing is strong, that drives the diapycnal diffusivity experienced by the tracer from $\mathcal{O}(10^{-5}) \text{ m}^2 \text{ s}^{-1}$ to $\mathcal{O}(10^{-4}) \text{ m}^2 \text{ s}^{-1}$. In particular, the long residence time around isolated pronounced topographic features including the Shackleton fracture zone, the Phoenix ridge, and the west Scotia Sea ridge system contributed most of the mixing across density surfaces that are typically 2,000 m above the bottom.

Larger estimates of mixing from tracer estimates than from microstructure profiles have been reported in other experiments. In the tracer release experiment in the Brazil Basin[5] tidal modulation of dissipation of kinetic energy was invoked to close the gap between the two estimates, but the argument was later challenged[34]. Our work suggests that lateral advection of tracer over rough topography where residence time is longer can provide a complementary explanation. A recent field programme in the equatorial Pacific Ocean found that over the rough topography of the Samoan Passage the microstructure-based diffusivity was a factor of 2 to 6 smaller than that inferred from a heat budget of the region, a bulk estimate similar in spirit to

tracer-based estimates[35]. While under-sampling of strong turbulence was proposed as an explanation for the large discrepancy, our work suggests an alternative or complementary explanation. Finally, it is worth mentioning that no discrepancy between microstructure-based and tracer-based diffusivities was reported in absence of rough topography in a tracer release experiment in the North Atlantic Ocean[36] as well as in the South-East Pacific sector in DIMES[11].

The suggestion that boundary-enhanced mixing affects the distribution of tracers over large areas in the ocean interior is not new. Armi[2,28] first suggested that the turbulent mixing within the benthic boundary layer, that is, the region with thickness of $\mathcal{O}(10)$ m in which density surfaces intersect boundaries, contributed most of the mixing of tracers in the abyssal and deep ocean. This signal would then be communicated to the rest of the ocean as fluid and tracers were advected away from the benthic boundary layers. This scenario was later refuted since the boundary layers are mostly unstratified and so cannot host efficient mixing[32]. In this study, we pointed out that the microstructure measurements indicate that the enhanced mixing above topography is not confined to the benthic boundary layers, rather it is the result of vigorous breaking of internal waves, which extends for several hundred metres above the benthic layer (Fig. 2c). Unlike in the benthic boundary layer, this region is well stratified (Fig. 2a) and the breaking waves drive very strong mixing rates.

Our study provides evidence that large mixing and long residence times of waters over rough topographic features extend their influence on tracer distributions over the rest of the stratified ocean. This may have important implications for our understanding of the overall role of diapycnal mixing in the ocean beyond the Drake Passage. Garabato et al.[13] show that diapycnal diffusivities must be of $\mathcal{O}(10^{-4})\,\mathrm{m^2\,s^{-1}}$ if they are to contribute a leading order transport of $\mathcal{O}(10)$ Sv in the MOC of the Southern Ocean below 2,000 m. Problematically, there are few seamounts and ridges across the Southern Ocean to substantially affect the area-averaged diapycnal diffusivity of the basin. This is exactly what we found in the Drake Passage region as well, where fixed depth and along-isopcynal averages of the microstructure inferred map of $\kappa$ were not very different from background values of $\mathcal{O}(10^{-5})\,\mathrm{m^2\,s^{-1}}$ at mid-depths. However, an accurate estimate of the mixing experienced by a tracer requires accurate knowledge of the residence time spent by the tracer close to the hotspots of strong mixing, which in turn depend on the frequency with which geostrophic eddies and mean flows bring the tracer close to topography and on the topographic steering of currents which tend to trap tracers close to topography. We speculate that a similar picture may apply to the whole Southern Ocean and possibly to all oceans. It should be a priority in future work to estimate the residence time of tracer close to rough topographic features, where mixing is strong, to finally answer the question of whether deep mixing is a leading order process below 2,000 m, where most seamounts and ridges are found[27,37].

## Methods

**Different weighting for $\kappa$.** In the main text, we defined $\bar{\kappa}$ as the value of average $\kappa$ weighted by the tracer concentration, arguing that this $\bar{\kappa}$ would represent the diapycnal mixing experienced by the tracer. One could equally well argue that weighting $\kappa$ by some power of $c$, like $c^2$, is equally reasonable. Taking a different line of reasoning, it could also be argued that $\kappa$ ought to represent the mean diapycnal diffusivity that drives the observed dissipation of tracer variance, $\chi$,

$$\chi = -6 < \kappa\, c_z^2 > \ \equiv\ -6\,\bar{\kappa}<c_z^2>. \tag{4}$$

This is tantamount to weighting the average $\kappa$ by $c_z^2$,

$$\bar{\kappa} \equiv \frac{<\kappa\, c_z^2>}{<c_z^2>}. \tag{5}$$

In Fig. 10, we compare these three definitions and show that they result in very comparable estimates of $\bar{\kappa}$ (to within a factor of 2).

**Data availability.** The tracer and hydrographic data used in the analysis are available through the online version of Watson et al.[17] The microstructure data can be obtained by contacting Louis St. Laurent and Alberto Naveira Garabato. The model setup and the surface and lateral boundary conditions are available from the corresponding author on reasonable request.

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

## Acknowledgements

We thank Glenn Flierl for enlightening discussions and three anonymous reviewers for insightful suggestions. Financial support for A.M. and R.F. under the US National Science Foundation grant OCE-1233832 is gratefully acknowledged. A.M. also acknowledges support from an NSERC PDF award.

## Author contributions

A.M. and R.F. designed the numerical experiment and the general approach, and guided the research. S.M., L.S.L. and A.N.G. provided and helped analysing the microstructure profiles. J.R.L. provided and helped analysing the tracer data. A.M. and R.F. drafted the article, and all authors contributed to the final manuscript.

## Additional information

**Competing financial interests:** The authors declare no competing financial interests.

