## [Peer Review File · Nature Communications]

Reviewers' comments:

Reviewer #1 (Remarks to the Author):

This study explores the relationship between local estimates of diffusivity and estimates from tracer release experiments, the latter being by nature an integrated (in space and time) value over the area occupied by the trace. This connection is key to understand the relevance of the highly heterogeneous maps of mixing rates for the large-scale transport of tracers and ultimately the global MOC.

In the present case (Drake Passage region), the paper manuscript makes a convincing argument that the heterogeneous distribution of mixing (here due to a decaying profile away from the bottom) explains the effective diffusivity experienced by a tracer advected by the meandering/turbulent ACC. Going into the mechanism, the authors show that this is not just a case of having enough high bathymetry "sticking out" into the thermocline. They suggest that water parcels tend to spend (on average) more time near topographic features. That is, the effective diffusivity of the tracer is heavily biased toward high value because of a slow down of currents close to the bottom.

This is an important point: the effective mixing of the tracer is not just a spatial average of diffusivity on an isopycnal surface. It is also a time-weighted average, factoring in the residence time of parcels near topography. This relies on a coupling with the circulation and how eddies/ACC interact with the bottom topography.

I find the paper very interesting and I enjoyed reading it, but this last point is where the paper is a bit frustrating. Although the suggested mechanism is attractive and plausible, the authors only show weak evidence to support it: the KE is small near topography. There is a jump between this piece of evidence and effectively showing that water parcels hang around longer near bottom features and areas of high diffusivity (one could easily think of counter examples where high KE does not imply short residence time, e.g. standing eddie locked to topography and strong recirculation).

In my opinion, this deserves a better demonstration. Could Lagrangian floats provide a more substantial diagnostic that indeed water parcels "stick" to topography? (technically, this is feasible, the MITgcm has a float package and hundredth of floats could be released at once to build up statistics).

Aside from this missing last step, the paper is of high quality; the topic is relevant to a wide community up to climate applications, the modeling and analysis tools are appropriate, figures are clear and the manuscript reads effortlessly.

I would happily support publication once a more robust proof of the long residence time near topography is provided.

Additional comments/questions:

1) First sentence of main text (page 1): "turbulent mixing plays a leading role in driving the MOC."

First sentence of abstract: the exact same statement "remains an open question".

You can't have it both ways. Is it sure or is it open?

2) Page 1, "Munk first estimated ... than observed." Page 1-2, "More complete inverse ... 2000 m".

I'm wondering if these statements are not in fact a source of confusion. Both Munk's and the inverse box model estimates are attached to a long list of assumptions and problems. Do we believe that Munk's estimate has anything to do with the effective tracer diffusivity in Drake passage?

3) page 3 "It further leads us to ... observed in the abyssal ocean". It is a huge stretch to make that statement. If anything Drake Passage is a very unusual place (a constricted passage where the most powerful current in the world has to go, with rough topography and a sill). The only similar place I can think of is the Indonesian Through Flow.

4) page 4: " the approach of assuming that the microstructure κ profiles are the "truth"."

"Truth" is not a good word in my opinion, it makes it sound as if the tracer release is incorrect. Similarly, in other places (e.g. abstract), the phrasing suggests there is something incorrect with one of the measurement techniques, or maybe that it is just a technical problem that needs to be solved.

Both types of measurement are correct, they measure different quantities, one is "Eulerian", the other is more Lagrangian and averages in space and time the Eulerian measurements.

5) bottom of page 4, the prescribed profile of $k(h_{ab})$: it would be worth to refer to to Fig. 7b at this point.

6) page 7: "It is verified (not shown here) ... vertical spreading of a numerical tracer."

Are you saying that you have verified that the spurious numerical mixing is below $10^{-5} \text{ m}^2/\text{s}$, i.e. small compared to the signal you care about here?

Typos:

page 1. line 9 from bottom: " fives decades"

page 8, before to last line : " due to to advection"

page 9, line 5, "tFig. 6"

figure 2, caption: " See Methods Section ..." did not find any "Methods" section.

Reviewer #2 (Remarks to the Author):

This is an interesting, generally well-written, and potentially important paper on ocean mixing, as it makes the point that *net* diapycnal mixing in the ocean appears to necessarily involve the rates at which water moves laterally and in a heaving motion. As it stands, the reader is left with some puzzles that ought to be at least noted in a general audience publication.

Why should flows be smaller near topography? Generally flow constrictions accelerate motions to conserve mass. Maybe there is some explanation in terms of the imposed boundary conditions?

The paper begins by invoking the Munk value from a one-dimensional advection-diffusion balance.

But it is not clear to me how to relate that nominal open ocean one-dimensional balance result to what has become a fully three dimensional distribution. Clarify? (Although usually interpreted as an open ocean result, Munk's profiles were close to the western and eastern boundary regions of the Pacific.)

More generally, is the Drake Passage unique or are the results here applicable to the global ocean?

Some small points:

P. 1. The idea of mixing "driving" a circulation is a bit odd. Perhaps "supporting". And the statement about an MOC 10 times larger would only be correct if the stratification magically stayed the same.

P. 3. I don't think anyone has ever "observed" the MOC--it has been calculated or estimated in different ways.

P. 4 Will a non-specialist reader have any idea what is meant by WOCE SR1b line ?

P. 5 There is discussion of lines not used, and which do not appear on the position chart. Why not drop this distracting bit?

P. 6 OCCA is not a climatology--it's an 18 month snapshot.

P. 12, 13. Wasn't Armi discussing a flat-bottom solution? Given the difference between mixing at topography and over an abyssal plain, the emphasis (which is repeated on P. 13) of the analogy seems forced and misleading (although again, experts will know that).

Fig. 2. The red dashed line in the legend has vanished.

Reviewer #3 (Remarks to the Author):

Review of NCOMMS-16-10058 (Mashayek et al)

This is an interesting paper with a few flaws that ought to be fixed before publishing. The purpose of the paper is to explain an apparent inconsistency between dye (integral) and microstructure (local) measurements of turbulence diffusion from a large scale mixing experiment in the Southern Ocean. The inconsistency is reconciled through use of a high-res model that incorporates data-based diffusivity profiles with higher mixing near the seafloor (wherever the seafloor is) assigned to the full domain. Model results indicate that lateral motions inject fluid parcels into near-bottom regions (I think mostly seamounts in this case) and returns them to the interior at a rate that enhances the integrated mixing of the fluid parcels so that the apparent diffusivity matches that of the dye (although not that of the interior mixing).

I like the approach, the analysis and the result. I would very much like to see it published.

That mixing near boundaries is enhanced has been the focus of much work in the past 25 y. The authors make an important contribution in reconciling the high dye diffusivity with the low bottom diffusivity in terms of two factors. The first is the rapid lateral stirring in this part of the ocean. The other is the extended residence times of fluid parcels near high-mixing regions at the bottom where currents are slower.

By the way, I think the authors should reword their abstract to emphasize these points - maybe more elegantly than I have stated it.

The end result is that the two disparate measurements are reconciled, just as they were (with less

effort) from the NATRE experiment.

The authors should work diligently to eliminate typos in the ms.

It is impossible to see anything of significance in Figure 5. I am guessing that what we see is simply the last data plotted from an extensive amount of model output. The highlights (near bottom mixing and residence times) are obscured by what is plotted on top? I don't think this figure helps and the paper doesn't need it.

I think that this study is related to previous work that has indicated the ejection of near-bottom fluid into the interior in the form of intermediate nepheloid layers (Thorpe et al, 1990; Moun et al 2002; Phillips et al, 1986; maybe early Armi). In these cases the mixing of fluid parcels in the interior was assigned to past bottom boundary layer residence rather than local mixing. It seems that the authors offer the same general prescription.

The statement in the abstract that the "model ... explains this seeming discrepancy" is too strong. After all, this is a model. Something more like "provides a scenario that is consistent with ..." would be more appropriate.

Reviewer #1 (Remarks to the Author):

This study explores the relationship between local estimates of diffusivity and estimates from tracer release experiments, the latter being by nature an integrated (in space and time) value over the area occupied by the trace. This connection is key to understand the relevance of the highly heterogeneous maps of mixing rates for the large-scale transport of tracers and ultimately the global MOC.

In the present case (Drake Passage region), the paper manuscript makes a convincing argument that the heterogeneous distribution of mixing (here due to a decaying profile away from the bottom) explains the effective diffusivity experienced by a tracer advected by the meandering/turbulent ACC. Going into the mechanism, the authors show that this is not just a case of having enough high bathymetry "sticking out" into the thermocline. They suggest that water parcels tend to spend (on average) more time near topographic features. That is, the effective diffusivity of the tracer is heavily biased toward high value because of a slow down of currents close to the bottom. This is an important point: the effective mixing of the tracer is not just a spatial average of diffusivity on an isopycnal surface. It is also a time-weighted average, factoring in the residence time of parcels near topography. This relies on a coupling with the circulation and how eddies/ACC interact with the bottom topography.

Re: We are glad that the reviewer found the topic and analysis of interest and importance. The above is a nice summary of the work that we take as reassuring evidence that the presentation of our results is clear.

I find the paper very interesting and I enjoyed reading it, but this last point is where the paper is a bit frustrating. Although the suggested mechanism is attractive and plausible, the authors only show weak evidence to support it: the KE is small near topography. There is a jump between this piece of evidence and effectively showing that water parcels hang around longer near bottom features and areas of high diffusivity (one could easily think of counter examples where high KE does not imply short residence time, e.g. standing eddies locked to topography and strong recirculation).

Re: We agree that the link between the slowdown of flow and the increase in residence time of tracer needed more clarification. We have addressed this point in the revised manuscript by performing new analysis and adding two additional figures, as we describe below. We believe that this clarification has made the paper stronger and we thank the reviewer for this comment.

In my opinion, this deserves a better demonstration. Could Lagrangian floats provide a more substantial diagnostic that indeed water parcels "stick" to topography? (technically, this is feasible, the MITgcm has a float package and hundredth of floats could be released at once to build up statistics).

Re: We think that our new analysis based on the evolution of the tracer achieves the same goal as an analysis based on Lagrangian floats and makes closer contact to the available observations. First, we realized that if the tracer does indeed spend more time close to topography, then one should see that tracer concentrations peak around seamounts and ridges. This is now shown in the new Fig. 9, which plots the column integrated tracer concentration on top of a bottom topography map. This

prediction could be tested in future observational campaigns designed to sample the tracer concentration at a higher resolution than done in DIMES. Second, we showed that the fraction of tracer within 1000 m of bottom topography is much higher than what one ought to find if the tracer was uniformly spread as a function of height above the bottom. This is illustrated in the new Fig. 10 and implies that tracer spends more time close to topography than elsewhere. In addition Fig. 10 shows that the excess of tracer lingering within 1000 m of topography accounts for the large net diffusivity that the tracer experiences in the domain. Together the two new figures provide visual and quantitative evidence that tracer spends more time close to rough topography than elsewhere and this "trapping" around topography leads to enhancement of net mixing experienced around the tracer.

Aside from this missing last step, the paper is of high quality; the topic is relevant to a wide community up to climate applications, the modeling and analysis tools are appropriate, figures are clear and the manuscript reads effortlessly.

Re: We appreciate the positive feedback and hope that our efforts have addressed the single issue raised.

I would happily support publication once a more robust proof of the long residence time near topography is provided.

Additional comments/questions:

1) First sentence of main text (page 1): "turbulent mixing plays a leading role in driving the MOC."
First sentence of abstract: the exact same statement "remains an open question".
You can't have it both ways. Is it sure or is it open?

Re: Fixed.

2) Page 1, "Munk first estimated ... than observed." Page 1-2, "More complete inverse 2000 m".

I'm wondering if these statements are not in fact a source of confusion. Both Munk's and the inverse box model estimates are attached to a long list of assumptions and problems. Do we believe that Munk's estimate has anything to do with the effective tracer diffusivity in Drake Passage?

Re: Munk's original estimate was for the Pacific subtropical gyre, so it is true that his original result has no direct bearing on the Southern Ocean. However the overall result that tracer distributions require large mixing rates at mid-depth (i.e. $\kappa \sim 1e-4 \text{ m}^2/\text{s}$) applies in the Southern Ocean as in the Pacific as shown for example in Naveira Garabato et al. (Nature, 2007). We modified the text to clarify in what sense we appeal to Munk's argument. Following the reviewer's remark, we have rewritten the last paragraph in the conclusion to focus on how our results may help quantify the mixing experienced by tracers in the deep ocean, rather than attempting to extrapolate our results to the global ocean in the footsteps of Munk.

3) page 3 "It further leads us to ... observed in the abyssal ocean". It is a huge stretch to make that statement. If anything Drake Passage is a very unusual place (a constricted passage where the most powerful current in the world has to go, with rough topography and a sill). The only similar place I can think of is the Indonesian Through Flow.

Re: As discussed above, we have modified the last paragraph of the conclusions to avoid over-extrapolation of our results. We explain that the main message of our paper is that in the Drake Passage region, the large net mixing at mid-depths is primarily influenced by the long residence time of tracers close to rough topography where mixing is strong. This result may be of relevance to the global ocean and explain why attempts to estimate the deep ocean mixing by simply averaging all mixing hot spots next to rough topographic features, without accounting for the increase residency time, come short of explaining why tracers seem to experience a net $\kappa \sim 1e-4 \text{ m}^2/\text{s}$.

4) page 4: " the approach of assuming that the microstructure κ profiles are the "truth"."

"Truth" is not a good word in my opinion, it makes it sound as if the tracer release is incorrect. Similarly, in other places (e.g. abstract), the phrasing suggests there is something incorrect with one of the measurement techniques, or maybe that it is just a technical problem that needs to be solved.

Both types of measurement are correct, they measure different quantities, one is "Eulerian", the other is more Lagrangian and averages in space and time the Eulerian measurements.

Re: We modified the language as suggested.

5) bottom of page 4, the prescribed profile of $k(h_{ab})$: it would be worth to refer to to Fig. 7b at this point.

Re: Done.

6) page 7: "It is verified (not shown here) ... vertical spreading of a numerical tracer."

Are you saying that you have verified that the spurious numerical mixing is below $10^{-5} \text{ m}^2/\text{s}$, i.e. small compared to the signal you care about here?

Re: Yes. We made the language more clear.

Typos:

page 1. line 9 from bottom: " fives decades"

page 8, before to last line : " due to to advection"

page 9, line 5, "tFig. 6"

figure 2, caption: " See Methods Section ..." did not find any "Methods" section.

Re: all fixed.

Reviewer #2 (Remarks to the Author):

This is an interesting, generally well-written, and potentially important paper on ocean mixing, as it makes the point that *net* diapycnal mixing in the ocean appears to necessarily involve the rates at which water moves laterally and in a heaving motion. As it stands, the reader is left with some puzzles that ought to be at least noted in a general audience publication.

Re: We appreciate the positive feedbacks. As we discuss below, we hope to have satisfactorily addressed the puzzles in the revised manuscript.

Why should flows be smaller near topography? Generally flow constrictions accelerate motions to conserve mass. Maybe there is some explanation in terms of the imposed boundary conditions?

Re: While it is true that at some constrictions there might be acceleration, in general flow speeds decay approaching bottom topography. In the Drake Passage, the ACC jets are known to have an equivalent barotropic structure with velocities decaying toward the bottom and data from FDRAKE moorings show this clearly. In the model, a bottom drag condition is imposed which leads to slowdown of flow in vicinity of topography.

The paper begins by invoking the Munk value from a one-dimensional advection-diffusion balance. But it is not clear to me how to relate that nominal open ocean one-dimensional balance result to what has become a fully three dimensional distribution. Clarify? (Although usually interpreted as an open ocean result, Munk's profiles were close to the western and eastern boundary regions of the Pacific.)

More generally, is the Drake Passage unique or are the results here applicable to the global ocean?

Re: There is a commonality between our study in the Drake Passage and mixing on a global scale. The discrepancy between the mixing required as net mixing at mid-depths to sustain the MOC (i.e. $\kappa \sim 1e-4 \text{ m}^2/\text{s}$) and what is typically measured with by microstructure probes at mid-depths ($\kappa \sim 1e-5 \text{ m}^2/\text{s}$) is a puzzle in the Drake Passage as much as in other ocean basins. The main message of our paper is that in the Drake Passage region, the intense mixing experienced by tracers at mid-depths is primarily set by the long residence time of tracers close to rough topography where mixing is enhanced. This result may be of relevance to the global ocean and explain why attempts to estimate the deep ocean mixing by simply averaging all mixing hot spots next to rough topographic features, without accounting for the increase residency time, come short of explaining why tracers seem to experience a net $\kappa \sim 1e-4 \text{ m}^2/\text{s}$. We have modified the discussion section (see last paragraph) to better describe this connection.

Some small points:

P. 1. The idea of mixing "driving" a circulation is a bit odd. Perhaps "supporting". And the statement about an MOC 10 times larger would only be correct if the stratification magically stayed the same.

Re: Language fixed. Thanks.

P. 3. I don't think anyone has ever "observed" the MOC--it has been calculated or estimated in different ways.

Re: Good point. We modified the text.

P. 4 Will a non-specialist reader have any idea what is meant by WOCE SR1b line ?

Re: It is now explained in the manuscript.

P. 6 OCCA is not a climatology--it's an 18 month snapshot.

Re: We now simply refer to it as a state estimate.

P. 12, 13. Wasn't Armi discussing a flat-bottom solution? Given the difference between mixing at topography and over an abyssal plain, the emphasis (which is repeated on P. 13) of the analogy seems forced and misleading (although again, experts will know that).

Re: Armi's argument was not limited to a flat-bottom. We feel that Armi's insight in the importance of boundary mixing deserves to be pointed out. But we also stress that there is a key difference between our conclusion and Armi's in that the mixing we talk about is due to internal waves in the stratified deep ocean, whereas Armi discussed focused on mixing within the unstratified boundary layer.

Fig. 2. The red dashed line in the legend has vanished.

Re: Oops. Fixed.

Reviewer #3 (Remarks to the Author):

Review of NCOMMS-16-10058 (Mashayek et al)

This is an interesting paper with a few flaws that ought to be fixed before publishing. The purpose of the paper is to explain an apparent inconsistency between dye (integral) and microstructure (local) measurements of turbulence diffusion from a large scale mixing experiment in the Southern Ocean. The inconsistency is reconciled through use of a high-res model that incorporates data-based diffusivity profiles with higher mixing near the seafloor (wherever the seafloor is) assigned to the full domain. Model results indicate that lateral motions inject fluid parcels into near-bottom regions (I think mostly seamounts in this case) and returns them to the interior at a rate that enhances the integrated mixing of the fluid parcels so that the apparent diffusivity matches that of the dye (although not that of the interior mixing).

I like the approach, the analysis and the result. I would very much like to see it published.

Re: We appreciate the positive feedback. As we discuss below, we have addressed the issues raised by the reviewer.

That mixing near boundaries is enhanced has been the focus of much work in the past 25 y. The authors make an important contribution in reconciling the high dye diffusivity with the low est diffusivity in terms of two factors. The first is the rapid lateral stirring in this part of the ocean. The other is the extended residence times of fluid parcels near high-mixing regions at the bottom where currents are slower.

By the way, I think the authors should reword their abstract to emphasize these points - maybe more elegantly than I have stated it.

Re: We thank the reviewer for pointing out the importance of our result and are glad the message has been clearly communicated. We have modified the abstract as well as the rest of the manuscript (including addition of new analysis) to better highlight the key importance of the residency time of tracers near rough topography.

The end result is that the two disparate measurements are reconciled, just as they were (with less effort) from the NATRE experiment.

Re: We think there is a fundamental difference between the NATRE and DIMES measurements. The tracer and microstructure-based measurements of mixing were consistent from the very beginning in NATRE, while there appeared to be an order of magnitude difference in DIMES. The dynamics and bathymetry of the Drake Passage gave rise to this seeming discrepancy in DIMES. In our view, the two measurements are not conflicting, but complementary as they are different methods to sample turbulence in time and space.

The authors should work diligently to eliminate typos in the ms.

Re: We did our best.

It is impossible to see anything of significance in Figure 5. I am guessing that what we see is simply the last data plotted from an extensive amount of model output. The highlights (near bottom mixing and residence times) are obscured by what is plotted on top? I don't think this figure helps and the paper doesn't need it.

Re: The manuscript and subsequent figures depend more on Fig. 5 in the revised manuscript. Specifically, Figs. 7, 8 and 9 now provide results that correspond to the specific time frame shown in Fig. 5 and so it is helpful for the reader to have a visual sense of tracer distribution at this specific snapshot. This is especially true for interpretation of the newly added Fig. 9. Thus, we decided to keep it in the manuscript. And is also visually appealing :)

I think that this study is related to previous work that has indicated the ejection of near-bottom fluid into the interior in the form of intermediate nepheloid layers (Thorpe et al, 1990; Moum et al 2002; Phillips et al, 1986; maybe early Armi). In these cases the mixing of fluid parcels in the interior was assigned to past bottom boundary layer residence rather than local mixing. It seems that the authors offer the same general prescription.

Re: The additional references have been added.

The statement in the abstract that the "model ... explains this seeming discrepancy" is too strong. After all, this is a model. Something more like "provides a scenario that is consistent with ..." would be more appropriate.

Re: We have toned down that the language.

REVIEWERS' COMMENTS:

Reviewer #1 (Remarks to the Author):

I am satisfied with the response to my comments.

In particular, the authors clarify the questions about the residence time/accumulation of tracers near topographic features.

I therefore recommend publication.

Minor comments:

- first line of section "Analysis ..": "we"  "We"
- line 259: "due to to advection"
- line 264: "tFig. 6"
- line 270: Fig. 6, do you mean Fig. 11 ?
- abstract: "residency time" (line 25 and 27). I admit English is to my first language, but "residency time" makes me think of a medical residency. "Residence time" sounds better in the present context.
Anyway, both terms are used and it maybe worth to settle for just one.

Reviewer #2 (Remarks to the Author):

The authors have addressed most (not all of my previous comments) and I believe the paper should be accepted as interesting and provocative. There do exist some generally small remaining issues they might want to consider.

The idea that mixing "drives" the circulation (lines 11,44) remains an odd one because mixing generally increases potential energy, not flow fields.

Not clear to me what they mean by "validating" (line 99). Testing? Calibrating? Comparing?

Line 207 Is a 50% error a "slight" overestimate?

The issue of why fluid lingers around topographic features is addressed, but not very convincingly. In a perfect Taylor column, the interior flow is stagnant, with the external flow going around and probably accelerating. In a stratified fluid, the column becomes a cone, and there is a literature permitting calculation of the exchange between the cone and the exterior. Is this irrelevant? Does it support the supposition? Perhaps time-dependence completely changes the behavior?

Fig. 2 the red dashed line remains undescribed.

Fig. 8a. Density surfaces and depth levels are two complete coordinate systems. Is it physically possible that the average of 1 is **always** larger than in the other? Puzzling.

CW

Reviewer #3 (Remarks to the Author):

This is better.

In terms of the relationship of NATRE to DIMES, yes the manner in which the fluid mixes differs. But there is no need to revise mixing physics (or efficiencies) to arrive at the same conclusion when considering dye diffusion and turbulence measurements. This bulk perspective is important.

I still don't know what Fig. 5 shows other than the last pixels drawn on the image. I don't think it shows what the authors would like which is the contrast between the high energy and low energy environments that the tracer experiences. I would exclude it from the paper.

I suggest that the authors consider these 2 points before acceptance.

RESPONSES REVIEWERS' COMMENTS:

Reviewer #1 (Remarks to the Author):

I am satisfied with the response to my comments. In particular, the authors clarify the questions about the residence time/accumulation of tracers near topographic features. I therefore recommend publication.

Re: We are glad that we were able to address the issues raised by the reviewer and thank them for their constructive comments.

Minor comments:

- first line of section "Analysis ..": "we"  "We"

Re: Fixed.

- line 259: "due to to advection"

Re: Fixed.

- line 264: "tFig. 6"

Re: Fixed.

- line 270: Fig. 6, do you mean Fig. 11 ?

Re: Yes. Fixed.

- abstract: "residency time" (line 25 and 27). I admit English is to my first language, but "residency time" makes me think of a medical residency. "Residence time" sounds better in the present context.

Anyway, both terms are used and it maybe worth to settle for just one.

Re: All changed to residence time.

Reviewer #2 (Remarks to the Author):

The authors have addressed most (not all of my previous comments) and I believe the paper should be accepted as interesting and provocative. There do exist some generally small remaining issues they might want to consider.

Re: We are glad that we were able to address most of the issues raised by the reviewer and thank them for their constructive comments. We addressed the remaining minor issues as discussed below.

The idea that mixing "drives" the circulation (lines 11,44) remains an odd one because mixing generally increases potential energy, not flow fields.

Re: Good point. We revised the text and eliminated the word "drive".

Not clear to me what they mean by "validating" (line 99). Testing? Calibrating? Comparing?

Re: The sentence has been changed to "...calibrating and testing the numerical simulations."

Line 207 Is a 50% error a "slight" overestimate?

Re: The word "slight" has been removed.

The issue of why fluid lingers around topographic features is addressed, but not very convincingly. In a perfect Taylor column, the interior flow is stagnant, with the external flow going around and probably accelerating. In a stratified fluid, the column becomes a cone, and there is a literature permitting calculation of the exchange between the cone and the exterior. Is this irrelevant? Does it support the supposition? Perhaps time-dependence completely changes the behavior?

Re: The reviewer is correct that the word "Taylor column" suggests a closed steady recirculation around a seamount. We have revised the text to point out that the topographically locked recirculation that we observe in the simulations are time dependent and thus exchange of waters between the recirculation and the exterior is possible. We also admit that more work is needed to understand these exchanges and how they set the residency time of waters around topographic features. This seems like a very important direction for future research, given its potential importance to quantify mixing rates in the global ocean.

Fig. 2 the red dashed line remains undescribed.

Re: fixed.

Fig. 8a. Density surfaces and depth levels are two complete coordinate systems. Is it physically possible that the average of 1 is *always* larger than in the other? Puzzling.

Re: The old Figure 7 showed the average of the diapycnal diffusivities over the portion of the isobaths and isopycnals where there was non negligible tracer concentrations. Thus the averages depended on how the tracer followed isobaths versus isopycnals and resulted in the puzzle raised by the reviewer. The question made us realize that taking the isobaths and isopycnal averages over the whole domain is better, because it removes the puzzle (the two averages are nearly identical) and conveys the same information. We eliminated the old Figure 7 and substituted with the new Figure 8, reproduced below.

Same as figure 7 (figure 8 in previous version of the manuscript) but now covering the whole domain and not just where tracer exists.

Reviewer #3 (Remarks to the Author):

This is better.

Re: Thank you.

In terms of the relationship of NATRE to DIMES, yes the manner in which the fluid mixes differs. But there is no need to revise mixing physics (or efficiencies) to arrive at the same conclusion when considering dye diffusion and turbulence measurements. This bulk perspective is important.

Re: We agree. The main difference between NATRE on one hand and DIMES and the Brazil Basin Tracer Release Experiment (BBTRE) on the other hand is that in NATRE the tracer was released at 300 m, way above any topography, while in the other two experiments the tracer was injected at depths deep enough for the tracer to encounter ridges and seamounts. Thus enhanced topographic mixing could affect the tracer-based estimates of mixing in DIMES and BBTRE, but not in NATRE. A discrepancy between tracer-based and microstructure-based estimates of mixing was reported in DIMES and BBTRE, but not in NATRE supporting our claim that the discrepancy is related to topographic enhanced mixing. We agree with the reviewer that this is an important point and we now discuss it in the conclusions.

I still don't know what Fig. 5 shows other than the last pixels drawn on the image. I don't think it shows what the authors would like which is the contrast between the high energy and low energy environments that the tracer experiences. I would exclude it from the paper.

Re: We think the figure is worth showing for the very reason raised by the reviewer to remove it. The figure shows that mixing is only enhanced in very few regions close to topography (the last few pixels in the figure), but these few regions dominate the overall mixing experienced by the tracer. We added text to the figure caption to highlight the importance of limited contact of tracer with bottom topography. Furthermore, we think that for a general audience it is helpful to have a visual illustration of the eddy field that stirs the tracer over the whole domain and advects it over topography; it also helps give a sense of the relative sizes of mesoscale eddies with respect to the domain extent. Last, but not least, this figure goes hand in hand with the important and newly added Fig. 7 in which the column-integrated tracer is projected on bathymetry to highlight the increase in residence time above rough topography.